# Impact of number and type of identified antigen on transplant-free survival in hypersensitivity pneumonitis

**Margaret Kypreos[1], Kiran Batra[2], Craig S. Glazer[1], Traci N. Adams[1]***

**1** Division of Pulmonary and Critical Care Medicine, University of Texas Southwestern Medical Center, Dallas, TX, United States of America, **2** Department of Radiology, University of Texas Southwestern Medical Center, Dallas, TX, United States of America

* traci.adams@utsouthwestern.edu

**Data Availability Statement:** All relevant data are within the paper and its Supporting Information files.

## Abstract

### Background

Identification of inciting antigen can affect diagnostic confidence, quality of life, and prognosis in patients with HP. It is unknown whether the number and type of antigen affect results of diagnostic testing or prognosis, whether antigen identified by clinical history alone affects prognosis, and whether feather exposure is associated with outcomes similar to those of other antigens.

### Methods

To evaluate whether the number or type of antigen identified by clinical history alone affects clinical outcomes, we evaluated a retrospective cohort of patients with a high or definite probability of HP based on recent guidelines.

### Results

In our retrospective cohort, 136 patients met high or definite probability of HP and were included in the analysis. Median transplant-free survival was better in patients with antigen identified on clinical history alone than patients without identified antigen. Feather exposure was associated with improved TFS compared to patients without antigen identified; there was no difference in TFS between patients with feather exposure and either mold or live bird exposure. Mold antigen was associated with increased risk of fibrotic HP compared to avian antigen. Among patients with identified antigen, the number and type of antigen did not affect TFS.

### Discussion

Our study suggests that clinical history is adequate for providing prognostic information to patients with HP and classifying the diagnostic probability of HP according to recent guidelines. Feather exposure should be considered an inciting antigen in patients with ILD.

**Funding:** The author(s) received no specific funding for this work.

**Competing interests:** The authors have declared that no competing interests exist.

## Introduction

Hypersensitivity pneumonitis (HP) is a group of granulomatous, interstitial, bronchiolar, and alveolar-filling pulmonary diseases caused by repeated exposure and sensitization to a variety of organic and chemical antigens [1]. Inciting antigens are typically microbial particulate matter such as mold or hot tubs, plant or animal proteins such as birds or feathers, or chemicals such as isocyanates [2]. Exposures may occur in the home, at work, or in recreational activities [3].

Identification of an inciting antigen is important for several reasons. Identifying potential antigens increases diagnostic confidence and is associated with improved quality of life and better prognosis [3–5]. Identification of antigen confirmed by an industrial hygienist or serum precipitating antibodies has been previously associated with improved survival in HP [4]. However, it remains unclear whether the number or type of identified antigens affects survival, radiographic or histopathologic findings and whether antigen identified on clinical history without confirmation by an industrial hygienist or the presence of serum precipitating antibodies is associated with survival in chronic HP.

To evaluate whether the number or type of antigen identified by clinical history alone affects clinical outcomes, we evaluated a retrospective cohort of patients with a high or definite probability of HP based on recent guidelines [3]. We also evaluated characteristics of high-resolution computed tomography (HRCT), transbronchial biopsy (Tbbx), bronchoalveolar lavage (BAL), and surgical lung biopsy (SLB) in these patients to determine whether the number or type of antigen affects diagnostic findings and potentially contributes to lead time bias. We hypothesized that identification of antigen exposure by the treating physician would be associated with improved transplant-free survival (TFS) but that the type of antigen or number of antigens would not be associated with TFS.

## Methods

We conducted a retrospective cohort study of all patients with a high or definite diagnosis of chronic HP based on recent guidelines [3]. We derived the HP cohort from the UT Southwestern pulmonary clinic Epic registry, which includes all patients seen in pulmonary clinic with a diagnosis of interstitial lung disease. Clinical data extracted from the medical record included age, gender, baseline pulmonary function testing, antigen exposure, BAL lymphocyte percentage, TBBx results, HRCT results, SLB results, survival, and transplant-free survival. We identified antigen exposure through a detailed history from a ILD specialist rather than a template questionnaire. An antigen was counted as avian if the patient was regularly exposed to a live bird or feather products. Mold exposure could be in the home or office or related to farming and was considered significant if the patient was regularly exposed to visible mold or regularly using a composte heap. An occupational medicine specialist (CSG) reviewed the exposure history in cases where it was unclear whether the exposure was significant enough to potentially lead to sensitization. Patients were classified as having 2 separate antigens if the category of identified antigen was different; for example, a feather comforter and a pet bird would be counted as avian antigen only, whereas home mold and a pet bird would count as 2 antigens. We defined a diagnostic BAL as a lymphocyte percentage greater than 30 [6]. HRCTs were reviewed by a thoracic radiologist (KB) who was blinded to the clinical diagnosis. We defined HRCT results as indeterminate, compatible, and typical HP based on recent guidelines [3]. The HRCT was defined as fibrotic or inflammatory based on the presence or absence of reticulations and traction bronchiectasis [3]. We defined TBBx and SLB results as typical, probable, or indeterminate for HP based on recent guidelines [3]. Based on the review of antigen exposure, HRCT, BAL, TBBx, and SLB, patients were classified as HP excluded, low probability

HP, moderate probability HP, high probability of HP, or HP based on recent guidelines. Only those with high or definite probability of HP were included in the study.

Continuous variables were expressed as means and standard deviations and comparisons were made using Student's t test. Categorical variables were expressed using counts and percentages; comparisons were made using Chi-squared test or Fisher's exact test, where appropriate. We used the Kaplan-Meier method to display and the log-rank test to compare survival curves. The association between antigen exposure and transplant-free survival were assessed using univariable and multivariable Cox proportional hazards regression. Known predictors of survival including age, previous smoking, presence of antigen, FVC % predicted, DLCO % predicted, and presence of fibrosis were included in a multivariable model [4].

The primary outcome of this study was transplant-free survival for patients with and without identified antigen exposure, defined as time from diagnosis of interstitial lung disease (ILD) to death or transplant. This study was approved by the Institutional Review Board at University of Texas Southwestern Medical Center (IRB approval protocol number STU-2021-0598), and consent was waived for the study.

## Results

In our retrospective cohort of 1157 patients with ILD, 136 patients met high or definite probability of HP and were included in the analysis. Demographic characteristics of the retrospective cohort are shown in Table 1. Mean age was 63 years, and 84% of the patients were non-Hispanic white. A potentially fibrogenic exposure was found in 84.6% of the cohort; of these, 88 patients (64.7%) had 1 antigen identified and 27 (19.9%) had more than 1 antigen identified. Baseline FVC in our cohort was 67.4% predicted, and baseline DLCO was 50.7% predicted. Sixty-eight percent of patients had an SLB performed for diagnosis, and 50% underwent bronchoscopy with Tbbx. All patients included in the study had a BAL, TBBx, and/or SLB for confirmation of diagnosis based on ATS criteria for high or definite HP. Median time from ILD diagnosis to death or transplant was 11.7 years.

TFS was significantly better in patients with antigen identified by detailed clinical history compared to those without identified antigen (median TFS 11.1 vs 4.9 years, respectively p = 0.003) (Fig 1). The presence of antigen was associated with TFS in a univariable model (HR 0.38, 95% CI 0.16–0.72, p = 0.005) of 136 patients. In a multivariable model with 134 patients (2 had missing DLCO baseline data and were excluded) adjusted for age, previous smoking, FVC % predicted, DLCO % predicted, and presence of fibrosis, identified antigen was associated with improved TFS (HR 0.39, 95% CI 0.17–0.89, p = 0.025) (Table 2). Because emphysema has been variably reported to impact mortality, the presence of emphysema, which was present in 25 patients (18.4%) in the cohort, was evaluated in a univariable model but did not impact TFS (HR 1.2, 95% CI 0.53–2.61, p = 0.59) so emphysema was not included in the multivariable model.

To evaluate whether the number of antigens identified was associated with outcome, TFS was compared between patients with no antigen, 1 antigen, and 2 or more antigens. TFS was significantly worse in patients with no antigen identified compared to those with either 1 antigen identified or 2 or more antigens identified by history, but TFS did not differ between patients with 1 antigen identified and those with 2 antigens identified (Fig 2). When estimating outcome risk, in a univariable model, the number of antigens was associated with TFS (HR 0.29, HR 0.31–0.95, p = 0.03), but in a multivariable model adjusted for age, smoking, FVC % predicted, DLCO % predicted, and fibrosis, the number of antigens was not associated with transplant-free survival (HR 0.31, 95% CI 0.31–1.05, p = 0.07).

**Table 1. Demographic characteristics of retrospective cohort.**

| | All HP (N = 136) | No antigen (N = 21) | 1 antigen (N = 88) | ≥2 antigens (N = 27) |
|---|---|---|---|---|
| Mean age (SD) | 63.0 (10.9) | 63.2 (7.7) | 61.9 (11.5) | 61 (10.5) |
| Male, No. (%) | 58 (42.6) | 11 (52.4) | 36 (40.9) | 10 (37.0) |
| Ethnicity, No. (%) | | | | |
| White | 115 (84.6) | 14 (66.7) | 78 (88.6) | 23 (85.2) |
| Black | 3 (2.2) | 1 (4.8) | 1 (11.4) | 1 (1.1) |
| Hispanic or Latino | 8 (5.9) | 2 (9.5) | 5 (5.7) | 1 (1.1) |
| Asian | 6 (4.4) | 4 (1.9) | 0 (0.0) | 2 (7.4) |
| Other | 0 (0.0) | 0 (0.0) | 0 (0.0) | 0 (0.0) |
| Unknown | 4 (2.9) | 0 (0.0) | 4 (4.5) | 0 (0.0) |
| Ever Smoker, No. (%) | 54 (39.7) | 7 (33.3) | 34 (38.6) | 13 (48.1) |
| Pack years, median (IQR) | 15 (6.4–24) | 21 (8–23) | 16.3 (8.1–24.5) | 10 (4–24) |
| Antigen identified, No. (%)* | 115 (84.6) | 0 (0.0) | 88 (100) | 27 (100) |
| Avian | 72 (52.9) | 0 (0.0) | 45 (51.1) | 27 (100) |
| Mold | 58 (42.6) | 0 (0.0) | 34 (38.6) | 24 (88.9) |
| Other** | 13 (9.6) | 0 (0.0) | 9 (10.2) | 4 (14.8) |
| Baseline Lung Function, mean (SD), N | | | | |
| FVC % predicted | 67.4 (19.3),134 | 63.3 (20), 20 | 68.7 (19.0),88 | 66.0 (19.0),26 |
| DLCO % predicted | 50.7 (17.4), 134 | 51.0 (22.5), 20 | 50.2 (17.0),88 | 52.2(13.6),26 |
| HRCT Available for Scoring, No. (%) | 136 (100) | 21 (100) | 88 (100) | 27 (100) |
| Lung Biopsy Performed, No. (%)*** | | | | |
| Surgical Biopsy | 93 (68.4) | 17 (81.0) | 58 (65.9) | 18 (66.7) |
| Transbronchial Biopsy | 68 (50) | 6 (28.5) | 45 (51.1) | 17 (63.0) |
| Bronchoalveolar Lavage**** | 57 (41.9) | 6 (28.5) | 39 (44.3) | 12 (44.4) |
| Follow Up Time in years, median (IQR) | 3.2 (1.8–5.5) | 1.83 (1.0–3.2) | 3.0 (1.9–5.5) | 2.8 (1.2–4.9) |
| Clinical Outcomes | | | | |
| Death, N (%) | 15 (11.0) | 4 (19.0) | 8 (9.1) | 3 (11.1) |
| Transplant, N (%) | 22 (16.2) | 5 (23.8) | 13 (14.8) | 4 (14.8) |
| Transplant-free survival, median years | 11.1 | 4.89 | 12.8 | 11.2 |

*1 patient had 3 antigens identified; the remainder had 2 antigens identified; antigen exposure was identified by history

**Other antigens included isocyanate exposure and fish tank exposure

***30 patients had both SLB and Tbbx

****49 patients had both BAL and TBBx

HRCT results were similar between patients with no identified antigen, 1 antigen, and 2 antigens (Table 3). There was no difference in the proportion of patients with indeterminate, compatible, and typical HP HRCT scans between groups. Sixteen percent of the cohort had inflammatory HP, and the proportion did not differ between groups (9.5% for no antigen, 18.2% for 1 antigen, and 14.5% for 2 or more antigens, p = 0.61). The proportion of patients with mosaicism, nodules, ground glass, and upper lobe predominance was not different between groups. When patients with no antigen were compared to those with any antigen identified, there was no difference in the proportion of patients with indeterminate, compatible, or typical HP scans, the proportion of patients with fibrotic vs inflammatory scans, or any of the inconsistent with UIP features. Histopathologic findings were also similar between

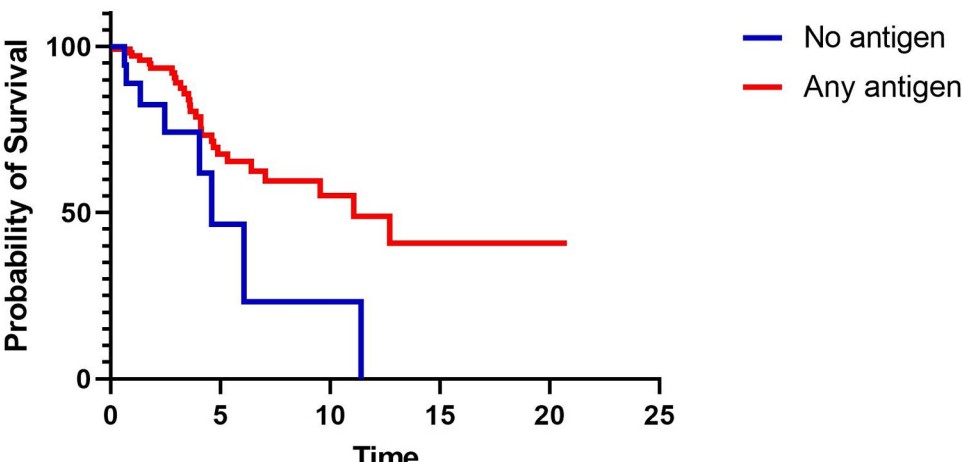

**Fig 1. Kaplan-Meier curve comparing transplant-free survival in years in patients with no antigen identified vs at least 1 antigen identified.**

patients with no identified antigen, 1 antigen, and 2 antigens (Table 4). There was no difference in the probability of HP based on Tbbx or SLB between patients with any antigen identified vs no antigen (p = 0.72 for Tbbx and p = 0.68 for SLB), and the proportion of patients undergoing Tbbx or SLB was also similar between groups.

To evaluate whether the type of antigen was associated with outcomes, TFS was compared between patients with no antigen identified (N = 21), feather antigen (N = 16), live bird antigen (N = 29), and mold antigen (N = 34). Patients with more than one identified antigen were not included this portion of the analysis in order to isolate the effect of each antigen. TFS was worst for patients with no identified antigen compared to those with feather, live bird, or mold antigen (Fig 3). When estimating outcome risk, patients with feather antigen exposure showed a decreased risk for death or transplant compared to patients with no antigen identified (HR 0.30, 95% CI 0.10–0.96, p = 0.043). Patients with feather antigen exposure had no difference in the risk of death or transplant compared to patients with live bird (HR 1.4, 95% CI 0.40–5.07, p = 0.59) or mold exposure (HR 0.70, 95% CI 0.15–3.2, p = 0.65). When patients with both live bird and feather exposure (all avian antigen) were compared to patients with mold antigen in a univariable model, there was no difference in TFS (HR 0.47, 95% CI 0.42–2.6, p = 0.91). Patients with avian antigen exposure were more likely than those with mold antigen exposure to have inflammatory HP (p = 0.014) and an HRCT that is inconsistent with UIP pattern (p = 0.01) (Table 5). The probability of HP based on HRCT, BAL lymphocyte percentage, and

**Table 2. Variables included in the Cox proportional hazards survival model.**

| Variable | HR for death or transplant | 95% CI | P value |
|---|---|---|---|
| Age | 1.011 | 0.98–1.05 | 0.54 |
| Previous smoking | 1.03 | 0.45–2.34 | 0.95 |
| Identified antigen | 0.39 | 0.17–0.89 | 0.025 |
| FVC % predicted | 0.98 | 0.96–1.01 | 0.22 |
| DLCO % predicted | 0.97 | 0.94–1.0 | 0.025 |
| Presence of fibrosis | 5.6 | 0.71–40.3 | 0.10 |

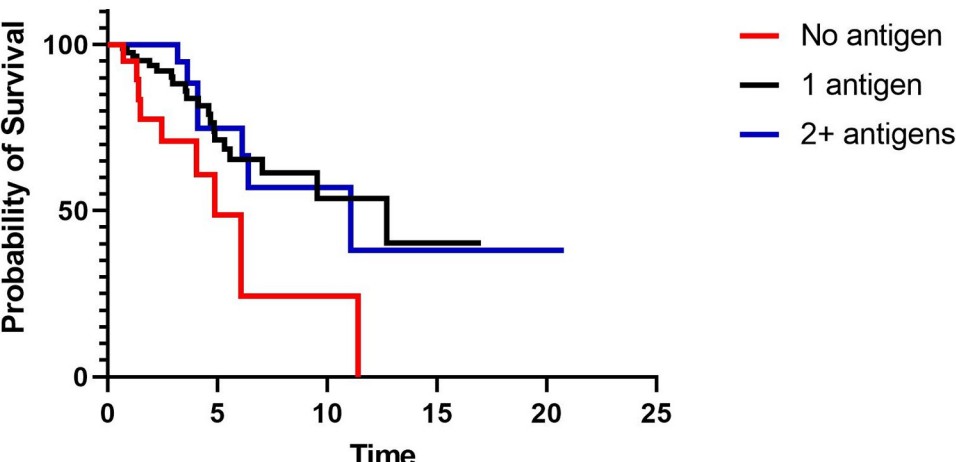

**Fig 2. Kaplan-Meier curve comparing transplant-free survival in years in patients with no antigen identified, 1 antigen identified, or 2 or more antigens identified.**

baseline PFTs were the same between groups. Patients with avian antigen exposure were more likely to undergo TBBx than patients with mold antigen, but among those who underwent TBBx the findings were not different between groups. The proportion of patients who underwent SLB and the probability of HP based on SLB results was similar between those with avian vs mold antigen.

**Table 3. Radiographic and histopathologic features of retrospective cohort.**

| Features | No antigen (N = 21) | 1 antigen (N = 88) | 2+ antigen (N = 27) | P value |
|---|---|---|---|---|
| HRCT pattern, N (%) | | | | 0.13 |
| Definite UIP | 1 (4.8) | 2 (2.3) | 0 (0) | |
| Possible UIP | 0 (0.0) | 10 (11.4) | 0 (0) | |
| Inconsistent with UIP | 20 (95.2) | 76 (86.4) | 27 (100) | |
| Type of HP | | | | 0.61 |
| Inflammatory | 2 (9.5) | 16 (18.2) | 4 (14.5) | |
| Fibrotic | 19 (90.5) | 72 (81.8) | 23 (85.2) | |
| HRCT honeycombing, N (%) | 5 (23.8) | 20 (22.7) | 8 (29.6) | 0.76 |
| Probability of HP based on HRCT, N (%)* | | | | 0.39 |
| Indeterminate | 7 (33.3) | 23 (26.1) | 3 (11.1) | |
| Compatible | 2 (9.5) | 12 (13.6) | 5 (18.5) | |
| Typical | 12 (57.1) | 53 (60.2) | 20 (74.1) | |
| Inconsistent features | | | | |
| Mid/upper lung predominant fibrosis | 7 (33.3) | 33 (37.5) | 6 (22.2) | 0.34 |
| Peribronchovascular predominance | 13 (61.9) | 36 (40.0) | 11 (40.7) | 0.20 |
| Extensive ground glass > reticulations | 11 (52.4) | 51 (58.0) | 16 (59.2) | 0.88 |
| Micronodules, No. (%) | 4 (19.0) | 19 (21.6) | 7 (25.9) | 0.84 |
| Mosaic attenuation in ≥ 3 lobes | 14 (66.7) | 62 (70.5) | 19 (70.4) | 0.94 |
| Cysts, No. (%) | 2 (9.5) | 10 (11.4) | 0 (0) | 0.19 |
| Consolidation, No. (%) | 1 (4.8) | 9 (10.2) | 0 (0) | 0.18 |

**Table 4. Transbronchial biopsy and SLB characteristics of retrospective cohort.**

| Histopathology features | No antigen (N = 21) | 1 antigen (N = 88) | 2+ antigen (N = 27) | P value |
|---|---|---|---|---|
| BAL performed | 6 (28.6) | 39 (44.3) | 12 (44.4) | 0.40 |
| BAL lymph > 30 | 5 (83.3) | 19 (21.6) | 5 (18.5) | 0.22 |
| Tbbx performed | 6 (28.6) | 45 (51.1) | 17 (63.0) | 0.06 |
| TBBx Findings | | | | |
| Indeterminate HP | 3 (50.0) | 17 (37.8) | 8 (47.1) | 0.72 |
| Probable HP | 0 (0) | 0 (0) | 0 (0) | |
| Typical HP | 3 (50.0) | 28 (62.2) | 9 (52.9) | |
| SLB performed | 17 (81.0) | 58 (65.9) | 18 (66.7) | 0.40 |
| SLB HP classification | | | | 0.68 |
| Indeterminate HP | 0 (0) | 3 (5.2) | 0 (0) | |
| Probable HP | 6 (35.3) | 17 (29.3) | 7 (38.9) | |
| Typical HP | 11 (64.7) | 38 (65.5) | 11 (61.1) | |
| SLB Findings | | | | |
| Poorly formed granulomas | 12 (70.6) | 37 (63.8) | 11 (61.1) | 0.83 |
| Airway-centered fibrosis | 16 (94.1) | 48 (54.5) | 18 (100) | 0.10 |
| Chronic fibrosing interstitial pneumonia | 15 (88.2) | 51 (87.9) | 18 (100) | 0.30 |
| Cellular interstitial pneumonia | 1 (5.9) | 7 (12.1) | 0 (0) | 0.25 |
| Cellular bronchiolitis | 1 (5.9) | 6 (10.3) | 0 (0) | 0.33 |

## Discussion

In this study, we examined the effect of number and type of antigen on transplant-free survival, radiographic, and histopathologic findings in a cohort of patients with a confident diagnosis of HP. Our study has several notable findings. 1) Identification of antigen by clinical history alone was associated with improved TFS. 2) Identification of a feather exposure was associated with improved TFS compared to unidentified antigen. 3) Among patients with identified antigen, the number and type of antigen did not affect transplant-free survival. 4) Patients with

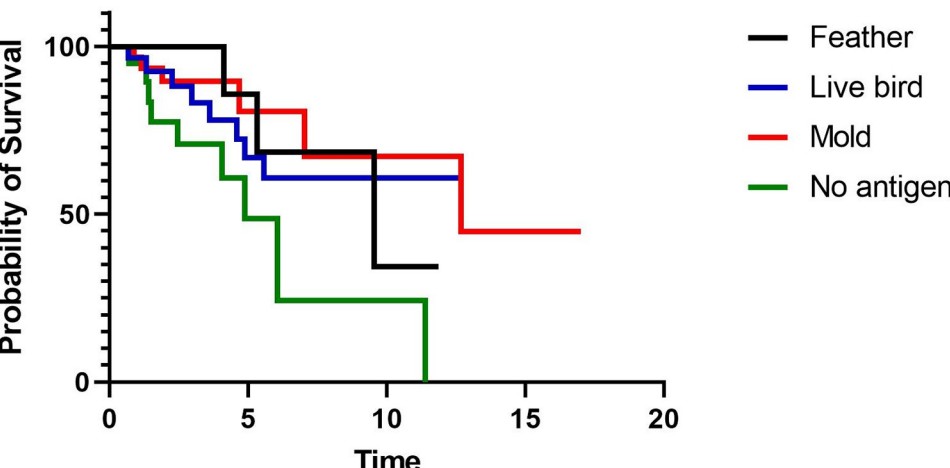

**Fig 3. Kaplan-meier curve comparing transplant-free survival in years in patients with no antigen, feather, live bird, or mold.**

**Table 5. Radiographic and histopathologic features by avian vs mold antigen.**

| | Avian antigen (N = 45) | Mold antigen (N = 34) | P-value |
|---|---|---|---|
| HRCT pattern | | | 0.01 |
| Definite UIP | 1 (2.2) | 0 (0) | |
| Possible UIP | 1 (2.2) | 8 (23.5) | |
| Inconsistent with UIP | 43 (95.6) | 26 (76.5) | |
| Type of HP | | | 0.005 |
| Inflammatory | 12 (26.7) | 1 (2.9) | |
| Fibrotic | 33 (73.3) | 33 (97.1) | |
| HRCT honeycombing | 7 (15.6) | 10 (29.4) | 0.14 |
| Probability of HP based on | | | 0.17 |
| HRCT | | | |
| Indeterminate | 8 (17.8) | 12 (35.3) | |
| Compatible | 6 (13.3) | 5 (14.7) | |
| Typical | 31 (68.9) | 17 (50) | |
| Inconsistent Features, N (%) | | | |
| Mid/upper lung predominance | 18 (40) | 12 (35.3) | 0.67 |
| Peribronchovascular | 21 (46.7) | 12 (35.3) | 0.22 |
| Extensive ground glass | 27 (60) | 18 (52.9) | 0.53 |
| Micronodules | 13 (28.9) | 5 (14.7) | 0.14 |
| Mosaic attenuation $\geq$ 3 lobes | 35 (77.8) | 21 (61.8) | 0.12 |
| Cysts | 5 (11.1) | 3 (8.8) | 0.74 |
| Consolidation | 6 (13.3) | 2 (5.9) | 0.28 |
| BAL performed | 22 (48.9) | 15 (44.1) | 0.67 |
| BAL lymph > 30 | 11 (50) | 6 (40) | 0.55 |
| BAL lymph mean (SD) | 35 (22.5) | 34.73 (31.1) | 0.98 |
| Tbbx performed | 29 (64.4) | 14 (41.2) | 0.04 |
| Tbbx pathology | | | |
| Indeterminate HP | 12 (41.4) | 4 (28.6) | 0.42 |
| Probable HP | 0 (0) | 0 (0) | |
| Typical HP | 17 (58.6) | 10 (71.4) | |
| SLB performed | 28 (62.2) | 22 (64.7) | 0.82 |
| SLB findings | | | 0.99 |
| Indeterminate HP | 1 (3.6) | 1 (4.5) | |
| Probable HP | 9 (32.1) | 7 (31.2) | |
| Typical HP | 18 (64.3) | 14 (63.4) | |
| Baseline FVC mean (SD) | 69.1 (18.1) | 69.1 (21.1) | 1.00 |
| Baseline DLCO mean (SD) | 52.7 (16.9) | 48.5 (16.8) | 0.28 |

mold exposure were more likely to have fibrosis on HRCT than patients with avian antigen exposure. 5) Emphysema is not a predictor of TFS in HP.

Our findings are similar to a prior study, which showed that identification of antigen confirmed by an industrial hygienist or serum precipitating antibodies was associated with improved survival in patients with biopsy-proven HP [4]. This may be explained by both a higher rate of fibrotic HP in patients without identified antigen and by reduced rate of FVC decline in patients who remove antigen compared to those who remain in exposure [7–12]. However, our study adds to the literature by demonstrating that the diagnostic evaluation, including radiographic and histopathologic probability of HP, do not differ between patients based on the identification of antigen. Our study indirectly supports the importance of antigen

removal by eliminating lead-time bias as a potentially contributing factor for improved survival in patients with identified antigen. We routinely recommend antigen remediation for all patients with identified antigen but are unable to rigorously assess whether that occurred in this retrospective study.

Our findings are also unique in that we demonstrated survival benefit with antigen identification alone without confirmation by an industrial hygienist or serum precipitating antibodies in patients with a confident diagnosis of HP and that we included patients with feather exposure alone [4, 10]. The use of an industrial hygienist may be limited by cost, availability, and expertise and lack of standardization of sample collection and analysis [13]. Further, while a positive serum precipitating antibody test may be informative that sensitization has occurred, available commercial tests have variable methods of measurement, different antigens, and nonuniform thresholds for positivity, which limit their negative predictive value [13]. Our study demonstrates that a thorough history for antigen exposure has prognostic value for patients with HP without the need for confirmatory testing. We agree with prior expert consensus statements that exposure history should be structured, standardized, and comprehensive [13], and several questionnaires have been previously published to guide exposure assessment [2, 14]. We suggest that exposure questionnaires also ascertain down exposure, as identification of down exposure in our study was associated with improved survival compared to patients with no antigen identified. While this study is the first to note survival association with down exposure compared to no identified antigen, it fits with prior HP studies that have included down as an inciting antigen and with numerous case series which have also suggested an association between down exposure and the development of HP [15–18].

The impact of the characteristics of antigen exposure, including type of antigen, intensity, and duration, on the development of hypersensitivity pneumonitis remains poorly understood [7]. Only a small percentage of people with antigen exposure will go on to develop HP, ranging from 8–540,000 per 100,000 per year among farmers and 6000–21000 per 100,000 per year in pigeon breeders [19–25]. What accounts for the difference in prevalence of disease in exposed patients and whether that translates to any meaningful difference in disease phenotype or outcome remains unclear. Our findings help to resolve conflicting data in the literature with regard to the effect of the type of antigen on mortality and fibrotic phenotype in HP. Two prior studies suggested that patients with avian antigen exposure have a better survival than patients with other types of exposure [4, 10], while other studies found that the type of antigen did not affect mortality [7, 26, 27]. We did not find an association between antigen type and TFS but did demonstrate that mold antigen leads to a higher proportion of fibrotic HP compared to avian antigen, a finding that has been shown in the literature and in our multivariate analysis of our cohort to lead to higher mortality [4, 10, 26]. Conversely, a prior study from the Mayo Clinic revealed that patients with avian antigen exposure were more likely to have fibrosis on HRCT than those without avian antigen, but this did not correspond to a change in mortality [26], while other studies did not find an association between antigen type and fibrotic phenotype [7, 27, 28]. Our study required high or definite diagnosis of HP, whereas the Mayo Clinic study only required histopathologic confirmation if antigen was not identified, which could have led to a lower confidence diagnosis and may account for different results.

Finally, our study elucidates the prevalence of emphysema in HP and its impact on TFS. Emphysema is a predictor of mortality in the literature in patients without other underlying lung disease [29], but the role that emphysema plays in mortality in patients with ILD remains unclear. In 2005, Cottin et al described the syndrome of combined pulmonary fibrosis and emphysema (CPFE), which is characterized by upper-lobe predominant emphysema and lower-lobe predominant fibrosis with preserved lung volumes and severely diminished DLCO

[30]. Studies comparing the mortality between CPFE and idiopathic pulmonary fibrosis have yielded conflicting results [31, 32]. However, in patients with scleroderma, mortality is higher in patients with both ILD and emphysema compared to patients with ILD without emphysema [33]. Our study adds to the literature by being the first to look for an association between emphysema and TFS in patients with HP, and like the prior studies of CPFE, we did not find an association between TFS and emphysema. It is possible that a confounder such as pulmonary hypertension, which is present in both CPFE patients and patients in scleroderma, may limit our ability to detect an association between emphysema and TFS.

Strengths of our study include the large cohort of patients with a highly confident diagnosis of HP, the identification of antigen by history alone, and the inclusion of down products as inciting antigen. The use of clinical history alone to identify antigen exposure increases the generalizability of our study. Further, we have limited selection bias by using published guidelines to define our HP cohort [3]. Finally, by analyzing feather exposure compared to no identifying antigen, we have strengthened the association between down exposure and the development of HP.

There are limitations to this study that should be acknowledged. The study is retrospective, and we were unable to determine for all patients if and when an antigen was removed. In addition, given the retrospective nature of the study, we could not accurately assess the influence of treatment at the time of HRCT or biopsy on the radiographic or histopathologic findings and could not control for lead-time bias other than adjustment for FVC and DLCO. We lacked statistical power to analyze fibrotic and non-fibrotic HP separately, though we did adjust for fibrosis in the multivariable model.

In summary, the identification of antigen by a clinician taking a detailed history is associated with improved transplant-free survival in HP but does not affect radiographic or histopathologic findings. Among those with identified antigen, the number of antigens does not affect transplant-free survival, radiographic, or histopathologic findings. The type of antigen does not influence transplant-free survival, but patients with mold antigen exposure are more likely to have fibrotic HP compared to those with avian antigen exposure. Patients with feather exposure had comparable transplant-free survival to those with live bird exposure but better transplant-free survival than patients without identified antigen. We suggest that clinical history is adequate for providing prognostic information to patients with HP and classifying the diagnostic probability of HP according to recent guidelines. Appropriate history taking and identification of antigen exposure is associated with prognosis of patients with ILD.

## Supporting information

**S1 Data.**
(XLSX)

## Author Contributions

**Conceptualization:** Margaret Kypreos, Kiran Batra, Craig S. Glazer, Traci N. Adams.

**Data curation:** Margaret Kypreos, Kiran Batra, Craig S. Glazer, Traci N. Adams.

**Formal analysis:** Margaret Kypreos, Traci N. Adams.

**Investigation:** Margaret Kypreos, Craig S. Glazer, Traci N. Adams.

**Methodology:** Margaret Kypreos, Craig S. Glazer, Traci N. Adams.

**Project administration:** Traci N. Adams.

**Resources:** Traci N. Adams.

**Validation:** Margaret Kypreos.

**Writing – original draft:** Margaret Kypreos, Kiran Batra, Craig S. Glazer, Traci N. Adams.

**Writing – review & editing:** Margaret Kypreos, Kiran Batra, Craig S. Glazer, Traci N. Adams.

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
