## [Decision Letter · Decision Letter 0]

1 Aug 2022

PONE-D-22-19345Impact of number and type of identified antigen on transplant-free survival in hypersensitivity pneumonitisPLOS ONE

Dear Dr. Traci Adams,

Thank you for submitting your manuscript to PLOS ONE. After careful consideration, we feel that it has merit but does not fully meet PLOS ONE’s publication criteria as it currently stands. Therefore, we invite you to submit a revised version of the manuscript that addresses the points raised during the review process. We appreciate the interesting study. However, there are some important points that are required to define clearly. Please carefully respond to the reviewers’ comments and suggestions. Please submit your revised manuscript by Sep 12 2022 11:59 PM. If you will need more time than this to complete your revisions, please reply to this message or contact the journal office at plosone@plos.org. Please include the following items when submitting your revised manuscript:A rebuttal letter that responds to each point raised by the academic editor and reviewer(s). You should upload this letter as a separate file labeled 'Response to Reviewers'.A marked-up copy of your manuscript that highlights changes made to the original version. You should upload this as a separate file labeled 'Revised Manuscript with Track Changes'.An unmarked version of your revised paper without tracked changes. You should upload this as a separate file labeled 'Manuscript'.

We look forward to receiving your revised manuscript.

Kind regards,

Vipa Thanachartwet, M.D.

Academic Editor

PLOS ONE

Journal Requirements:

Reviewers' comments:

Reviewer's Responses to Questions

**Comments to the Author**

1. Is the manuscript technically sound, and do the data support the conclusions?

Reviewer #1: Yes

Reviewer #2: Partly

2. Has the statistical analysis been performed appropriately and rigorously? 

Reviewer #1: Yes

Reviewer #2: No

3. Have the authors made all data underlying the findings in their manuscript fully available?

Reviewer #1: Yes

Reviewer #2: Yes

4. Is the manuscript presented in an intelligible fashion and written in standard English?

Reviewer #1: Yes

Reviewer #2: Yes

5. Review Comments to the Author

Reviewer #1: Kypreos et al. investigate the relationship between antigen type and transplant-free survival (TFS) in patients with hypersensitivity pneumonitis. The authors determined that identification of inciting antigen led to improved TFS. While TFS was not different in patients with mold or feather exposure, mold antigen was associated with increased risk of fibrotic HP. Additionally, in patients with an identified antigen, the number of antigens or the type of antigen did not affect TFS. The authors present interesting data, summarized below are my concerns.

1. Does the type of antigen exposure correlate with lung function (FVC or DLCO)?

2. Does the type of antigen exposure correlate with BAL lymphocyte percentage?

Reviewer #2: The study sought to determine if patients with HP with identifiable exposure antigens had better transplant free survival. This was a secondary database analysis that tested a hypothesis that had been previously confirmed by a more rigorous method of determining antigen exposure. The authors argued that the contribution of this study was that clinical history alone could be sufficient. However, the description in the methods including patient population, exposure history and statistical analysis was inadequate to allow for assessment of the validity of the results. Specific comments are outlined below.

1) How were the 136 cases of HP identified? Did these cases have HP diagnosis already or some numbers of patients with ILD diagnosis were matched with the guidelines and only those with high confidence or definitive HP were selected? Were they identified because they all had typical CT findings of small airway disease? A detailed description on how the cohort was established is critical to avoid any case selection bias. A flow chart on the patient selection process would help.

2) How were the antigen exposures identified and confirmed to be relevant? Based on review of the clinic notes, serum IgG tests, or other environmental tests (like indoor air quality test)? If the information was obtained from clinic notes, were these notes from pulmonologists/allergists? It is well known that adequate history-taking in HP requires a high index of suspicion, time, and content expertise. Differences in approach and content among clinicians lead to varying reliability.

3) HRCT was read by one of the co-authors. Why were the original CT reports not used? Potential bias could be introduced if the readers know the patient has a high pre-test probability of HP.

4) The guidelines were not published until 2020. Many of these patients likely were diagnosed with HP before 2020. What was the time frame from which the patients came from?

5) There were another HP guidelines published by CHEST in 2020. The guidelines prioritize environmental exposure in the algorithm. Were the results the same if the CHEST guidelines were used?

6) Page 5. "The primary outcome of this study was transplant-free survival... defined as time from diagnosis of interstitial lung disease (ILD) to death or transplant". The clinical course of HP is highly variable, especially between non-fibrotic and fibrotic HP. How did you account for lead time bias, especially for fibrotic HP? Also was the worse survival in HP patients without identifiable exposure due to fibrotic HP patients since many more had no identifiable exposure compared to non-fibrotic HP (Table 3)? If the analysis were done separately for fibrotic and non-fibrotic HP, would the same results be expected?

7) Line 80. Please describe how mold exposure was considered significant and relevant, since molds are ubiquitous in the environment.

8) Table 1. Please define "sensitizing antigens". Were they from history, IgG test or other methods?

9) Line 94. In the model, how were the missing values handled, since not all patients had PFT, lung biopsy etc?

10) In Table 1, in the 37 death and transplant, how many were fibrotic HP and how many were non-fibrotic HP? It was a bit of a surprise that in Table 2, presence of fibrosis was not a factor.

11) In Table 1 under antigen identified, what does "other" indicate?

12) Table 3. In the ATS guidelines, BAL and lung biopsy play a significant role in determining the diagnostic confidence. So without BAL or lung pathology, it is difficult to reach high confidence or definitive diagnosis of HP, especially for non-fibrotic HP. This could potentially introduce selection bias. This is a significant issue that needs to be addressed. Perhaps additional analysis should be performed only in patients who had lung biopsy and BAL.

6. PLOS authors have the option to publish the peer review history of their article (what does this mean?). If published, this will include your full peer review and any attached files.

Reviewer #1: No

Reviewer #2: No

---

## [Author Response · Author response to Decision Letter 0]

3 Aug 2022

Reviewer #1: Kypreos et al. investigate the relationship between antigen type and transplant-free survival (TFS) in patients with hypersensitivity pneumonitis. The authors determined that identification of inciting antigen led to improved TFS. While TFS was not different in patients with mold or feather exposure, mold antigen was associated with increased risk of fibrotic HP. Additionally, in patients with an identified antigen, the number of antigens or the type of antigen did not affect TFS. The authors present interesting data, summarized below are my concerns.

1. Does the type of antigen exposure correlate with lung function (FVC or DLCO)?

The type of exposure does not correlate with FVC or DLCO. We have added a sentence in the results section to note this and have included PFT data data in Table 5.

“The probability of HP based on HRCT, BAL lymphocyte percentage, and baseline PFTs were the same between groups.” 

2. Does the type of antigen exposure correlate with BAL lymphocyte percentage?

The type of exposure does not correlate with BAL lymphocyte percentage. We have added a sentence in the results section to note this and have included BAL data in Table 5.

“The probability of HP based on HRCT, BAL lymphocyte percentage, and baseline PFTs were the same between groups.” 

Reviewer #2: The study sought to determine if patients with HP with identifiable exposure antigens had better transplant free survival. This was a secondary database analysis that tested a hypothesis that had been previously confirmed by a more rigorous method of determining antigen exposure. The authors argued that the contribution of this study was that clinical history alone could be sufficient. However, the description in the methods including patient population, exposure history and statistical analysis was inadequate to allow for assessment of the validity of the results. Specific comments are outlined below.

1) How were the 136 cases of HP identified? Did these cases have HP diagnosis already or some numbers of patients with ILD diagnosis were matched with the guidelines and only those with high confidence or definitive HP were selected? Were they identified because they all had typical CT findings of small airway disease? A detailed description on how the cohort was established is critical to avoid any case selection bias. A flow chart on the patient selection process would help.

We identified the cases from our retrospective cohort of patients diagnosed with interstitial lung disease from 2005-2021. The cohort is derived from a database in our electronic medical record, which includes all patients with a diagnosis of interstitial lung disease (or any subtype of ILD including IPF, HP, etc) on the problem list who had been seen in the pulmonary clinic at UT Southwestern. Our cohort contains 1157 patients. Of these, 92 had moderate probability of HP by ATS criteria and were excluded. 136 patients had a high or definite probability of HP by ATS criteria and were included. Each of these patient’s medical records was reviewed, and our thoracic radiologist entered interpretations of each of the HRCT scans into a spreadsheet without knowledge of the diagnosis because she opens the imaging software only, rather than the full medical record where the diagnosis might be visible. Pathology findings, demographics, and antigen exposure were also entered into a spreadsheet, though we were not able to be blind to the diagnosis when entering data from the medical record. Once all available data was present for review, we used the objective data to determine whether the patient met ATS criteria and used a multidisciplinary discussion to confirm a diagnosis of HP if one had not been conducted previously. Most, but not all, of these patients had a prior diagnosis of HP in the medical record. We chose to only include those with a high or definite confidence level because to reach this level of confidence, ancillary information including a BAL or lung biopsy are needed, and we thought that inclusion of patients with moderate probability could introduce significant selection bias as those patients are not required to undergo bronchoscopy or lung biopsy.

We did not include a flow chart (though we can if the reviewer needs this), but have included the following statements in the manuscript:

“We derived the HP cohort from the UT Southwestern pulmonary clinic Epic registry, which includes all patients seen in pulmonary clinic with a diagnosis of interstitial lung disease.”

“Based on the review of antigen exposure, HRCT, BAL, TBBx, and SLB, patients were classified as HP excluded, low probability HP, moderate probability HP, high probability of HP, or HP based on recent guidelines. Only those with high or definite probability of HP were included in the study.”

“In our retrospective cohort of 1157 patients with ILD, 136 patients met high or definite probability of HP and were included in the analysis.”

2) How were the antigen exposures identified and confirmed to be relevant? Based on review of the clinic notes, serum IgG tests, or other environmental tests (like indoor air quality test)? If the information was obtained from clinic notes, were these notes from pulmonologists/allergists? It is well known that adequate history-taking in HP requires a high index of suspicion, time, and content expertise. Differences in approach and content among clinicians lead to varying reliability.

Each of these patients had been seen by an interstitial lung disease specialist at UT Southwestern. Our providers take a detailed exposure history and typically provide commentary about whether this exposure is considered significant. The majority of our providers use the following list of questions, which we did not include because it has not uniformly been used by our providers across the 16 years of the registry.

Type of home

Pier and beam or concrete slab

Crawl space inspection (wet, dry, mold, no mold)

Water damage

Visible mold in home

Type of air conditioning unit

Visible mold in air conditioning unit

Use of humidifier, pool, hot tub, sauna, CPAP/bipap

Bird exposure

Feather products in the home

Hobbies (gardening, metal/wood working, wind instrument)

We agree that it can be difficult to determine whether an exposure is relevant in some cases, as there is not a defined threshold that puts a patient at risk for HP. When there was any discrepancy about the exposure, Craig Glazer, who completed an occupational lung disease fellowship at National Jewish, reviewed the exposure to determine whether it was significant. We have added the following sentences to the Methods section to address this:

“Mold exposure could be in the home or office or related to farming and was considered significant if the patient was regularly exposed to visible mold or regularly using a composte heap. An occupational medicine specialist (CSG) reviewed the exposure history in cases where it was unclear whether the exposure was significant enough to potentially lead to sensitization.”

3) HRCT was read by one of the co-authors. Why were the original CT reports not used? Potential bias could be introduced if the readers know the patient has a high pre-test probability of HP.

We have only one thoracic radiologist at UT Southwestern, who has been involved in a number of studies and whose work has very closely correlated with radiologists at other centers including UC Davis, University of Chicago, and UCSF in prior studies. She read the scans without knowledge of diagnosis and did not know the exposure history or clinical diagnosis. This led to less variability than using the original CT reports, which may have been read by a radiologist without expertise in thoracic imaging.

 We have included a statement about this in the methods section: “HRCTs were reviewed by a thoracic radiologist (KB) who was blinded to the clinical diagnosis.”

4) The guidelines were not published until 2020. Many of these patients likely were diagnosed with HP before 2020. What was the time frame from which the patients came from?

Our electronic medical record began in 2005, so patients were included who were seen in our ILD clinic after 2005 through 2020. Our assignment of diagnosis was retrospective using the ATS guidelines as outlined under question 1. We included the following in the methods section.

“Based on the review of antigen exposure, HRCT, BAL, TBBx, and SLB, patients were classified as HP excluded, low probability HP, moderate probability HP, high probability of HP, or HP based on recent guidelines. Only those with high or definite probability of HP were included in the study.”

5) There were another HP guidelines published by CHEST in 2020. The guidelines prioritize environmental exposure in the algorithm. Were the results the same if the CHEST guidelines were used?

The Chest criteria do not overlap entirely with ATS criteria, as they classify antigen exposure as unidentified, indeterminate, and identified. Further, Chest guidelines allow for a confident diagnosis based on exposure and HRCT alone, which would have led to additional patients being included that are not included in high/definite classification of ATS criteria. However, patients who meet a high or definite confidence of HP diagnosis in the ATS guidelines also meet a “provisional high confidence” or “HP” category of diagnosis in the Chest guidelines, so all of our patients would have been included by Chest criteria as well. We felt that having a cohort that is better-defined with either BAL or biopsy as confirmation would lead to clearer conclusions than including patients who were defined only by antigen exposure and CT scan. We agree that inclusion of those patients who did not have bronchoscopy or lung biopsy could lead to selection bias.

We included the following statement in the results section: “All patients included in the study had a BAL, TBBx, and/or SLB for confirmation of diagnosis based on ATS criteria for high or definite HP.”

6) Page 5. "The primary outcome of this study was transplant-free survival... defined as time from diagnosis of interstitial lung disease (ILD) to death or transplant". The clinical course of HP is highly variable, especially between non-fibrotic and fibrotic HP. How did you account for lead time bias, especially for fibrotic HP? Also was the worse survival in HP patients without identifiable exposure due to fibrotic HP patients since many more had no identifiable exposure compared to non-fibrotic HP (Table 3)? If the analysis were done separately for fibrotic and non-fibrotic HP, would the same results be expected?

We are not able to account for lead-time bias which is consistent with other retrospective cohort studies in ILD, including the initial study by Fernandes-Perez. We try to adjust for that by adjusting for FVC and DLCO at the time of ILD diagnosis, but there is not a way to directly adjust for lead-time bias using available data from the medical record. We added the following statement to the discussion: “In addition, given the retrospective nature of the study, we could not accurately assess the influence of treatment at the time of HRCT or biopsy on the radiographic or histopathologic findings and could not control for lead-time bias other than adjustment for FVC and DLCO.”

If we did a separate analysis for fibrotic and non-fibrotic HP, we would be unlikely to have statistical power to demonstrate any conclusions given the number of variables we need to control for. We did control for fibrotic disease in our model, which is the best we can do to account for the differential finding. We included the following statement in the discussion: “We lacked statistical power to analyze fibrotic and non-fibrotic HP separately, though we did adjust for fibrosis in the multivariable model.”

7) Line 80. Please describe how mold exposure was considered significant and relevant, since molds are ubiquitous in the environment.

Molds are ubiquitous, but they only grow in moist environments. We count a mold exposure as significant if the patient was regularly exposed to visible mold or regularly using composte. Industrial hygienists would typically examine for visible mold and quantify indoor vs outdoor air quality, and if there is either visible mold or the counts are 10x indoors what they are outdoors that would count as a significant exposure. We used visible mold as a significant and relevant exposure. Any questionable exposures were reviewed by Craig Glazer. We added the following sentences to the methods section: “Mold exposure could be in the home or office or related to farming and was considered significant if the patient was regularly exposed to visible mold or regularly using a composte heap. An occupational medicine specialist (CSG) reviewed the exposure history in cases where it was unclear whether the exposure was significant enough to potentially lead to sensitization.”

8) Table 1. Please define "sensitizing antigens". Were they from history, IgG test or other methods?

We have clarified that these were antigens from history in the table. We also removed “Sensitizing antigen” from the text to avoid confusion and substituted: “A potentially fibrogenic exposure”

9) Line 94. In the model, how were the missing values handled, since not all patients had PFT, lung biopsy etc?

All variables in the univariate and multivariate analysis had complete data except for 2 patients who were missing DLCO. For univariable analysis, any patient missing data is excluded. In the multivariable analyses, any patient that is missing any of the included variables is excluded. We have added a statement in the manuscript about how many patients were included in this analysis. “The presence of antigen was associated with TFS in a univariable model (HR 0.38, 95% CI 0.16-0.72, p=0.005) of 136 patients. In a multivariable model with 134 patients (2 had missing DLCO baseline data and were excluded) adjusted for age, previous smoking, FVC % predicted, DLCO % predicted, and presence of fibrosis, identified antigen was associated with improved TFS (HR 0.39, 95% CI 0.17-0.89, p=0.025) (Table 2).”

10) In Table 1, in the 37 death and transplant, how many were fibrotic HP and how many were non-fibrotic HP? It was a bit of a surprise that in Table 2, presence of fibrosis was not a factor.

We will clarify this in the manuscript. Only 1 patient of the 37 patients who died was nonfibrotic. We suspect that it is the low number of nonfibrotic patients or the association between nonfibrotic disease and other included variables that led to the lack of statistical significance in the multivariable model. We have included the following statement in the discussion section: “We lacked statistical power to analyze fibrotic and non-fibrotic HP separately, though we did adjust for fibrosis in the multivariable model.”

11) In Table 1 under antigen identified, what does "other" indicate?

Other indicated antigen other than avian or mold. Our “other” cohort consisted of isocyanate exposure and fish tank exposure. We have included a note about this under Table 1: **Other antigens included isocyanate exposure and fish tank exposure

12) Table 3. In the ATS guidelines, BAL and lung biopsy play a significant role in determining the diagnostic confidence. So without BAL or lung pathology, it is difficult to reach high confidence or definitive diagnosis of HP, especially for non-fibrotic HP. This could potentially introduce selection bias. This is a significant issue that needs to be addressed. Perhaps additional analysis should be performed only in patients who had lung biopsy and BAL.

We only included patients with a high or definite confidence of HP, so all patients did undergo bronchoscopy or lung biopsy. The goal of including only these patients, rather than those who met a diagnosis of HP with moderate confidence by exposure + HRCT, is to minimize selection bias. We absolutely agree that including patients who did not have bronchoscopy or biopsy demonstrated HP would introduce selection bias. We included this statement in the results section to clarify: “All patients included in the study had a BAL, TBBx, and/or SLB for confirmation of diagnosis based on ATS criteria for high or definite HP.”

---

## [Decision Letter · Decision Letter 1]

9 Aug 2022

PONE-D-22-19345R1Impact of number and type of identified antigen on transplant-free survival in hypersensitivity pneumonitisPLOS ONE

Dear Dr. Adams,

Thank you for submitting your manuscript to PLOS ONE. After careful consideration, we feel that it has merit but does not fully meet PLOS ONE’s publication criteria as it currently stands. Therefore, we invite you to submit a revised version of the manuscript that addresses the points raised during the review process.

We appreciate your efforts for the study and the authors have made a careful revision to the manuscript. However, there are some minor points that are required to define clearly. Please carefully respond to the reviewer’ comments and suggestions. 

We look forward to receiving your revised manuscript.

Kind regards,

Vipa Thanachartwet, M.D.

Academic Editor

PLOS ONE

Journal Requirements:

Reviewers' comments:

Reviewer's Responses to Questions

**Comments to the Author**

1. If the authors have adequately addressed your comments raised in a previous round of review and you feel that this manuscript is now acceptable for publication, you may indicate that here to bypass the “Comments to the Author” section, enter your conflict of interest statement in the “Confidential to Editor” section, and submit your "Accept" recommendation.

Reviewer #1: All comments have been addressed

Reviewer #2: (No Response)

2. Is the manuscript technically sound, and do the data support the conclusions?

Reviewer #1: Yes

Reviewer #2: Yes

3. Has the statistical analysis been performed appropriately and rigorously? 

Reviewer #1: Yes

Reviewer #2: Yes

4. Have the authors made all data underlying the findings in their manuscript fully available?

Reviewer #1: Yes

Reviewer #2: Yes

5. Is the manuscript presented in an intelligible fashion and written in standard English?

Reviewer #1: Yes

Reviewer #2: Yes

6. Review Comments to the Author

Reviewer #1: The authors have addressed all of this reviewer's concerns. The manuscript is greatly improved. Great job!

Reviewer #2: The manuscript is now more clear. The response is satisfactory. Just a few remaining points.

1) Your response indicates that all patients included in the study had a BAL, TBBx, and/or SLB. I assume there were patients with BAL but no Tbbx? If so, please update Table 1 to include total number of patients with BAL (with or without Tbbx) in each column since BAL is a separate criterion in the diagnostic algorithm.

2) Please add IRB approval protocol number.

7. PLOS authors have the option to publish the peer review history of their article (what does this mean?). If published, this will include your full peer review and any attached files.

Reviewer #1: No

Reviewer #2: No

---

## [Author Response · Author response to Decision Letter 1]

9 Aug 2022

Reviewer #1: The authors have addressed all of this reviewer's concerns. The manuscript is greatly improved. Great job!

We appreciate the reviewer feedback.

Reviewer #2: The manuscript is now more clear. The response is satisfactory. Just a few remaining points.

1) Your response indicates that all patients included in the study had a BAL, TBBx, and/or SLB. I assume there were patients with BAL but no Tbbx? If so, please update Table 1 to include total number of patients with BAL (with or without Tbbx) in each column since BAL is a separate criterion in the diagnostic algorithm.

We have added the total number of patients with BAL (N=57, 49 of whom also had TBBx) to table 1 as requested.

We also created a note underneath the table with the following text to improve transparency with regard to diagnostic workup in our cohort.

***30 patients had both SLB and Tbbx

****49 patients had both BAL and TBBx

2) Please add IRB approval protocol number

We have added the IRB approval number.

“This study was approved by the Institutional Review Board at University of Texas Southwestern Medical Center (IRB approval protocol number STU-2021-0598), and consent was waived for the study.”

---

## [Editor Report · Decision Letter 2]

11 Aug 2022

Impact of number and type of identified antigen on transplant-free survival in hypersensitivity pneumonitis

PONE-D-22-19345R2

Dear Dr. Traci Adams,

We’re pleased to inform you that your manuscript has been judged scientifically suitable for publication and will be formally accepted for publication once it meets all outstanding technical requirements.

Kind regards,

Vipa Thanachartwet, M.D.

Academic Editor

PLOS ONE

Additional Editor Comments (optional):

All issues were revised according to the reviewers' comments and suggestions.
---

## [Editor Report · Acceptance letter]

23 Aug 2022

PONE-D-22-19345R2 

Impact of number and type of identified antigen on transplant-free survival in hypersensitivity pneumonitis 

Dear Dr. Adams:

I'm pleased to inform you that your manuscript has been deemed suitable for publication in PLOS ONE. Congratulations! Your manuscript is now with our production department. 

Kind regards, 

on behalf of

Associate Professor Vipa Thanachartwet 

Academic Editor

PLOS ONE